# Mid-Level Data Fusion Combined with the Fingerprint Region for Classification DON Levels Defect of Fusarium Head Blight Wheat

**DOI:** 10.3390/s23146600

**Published:** 2023-07-22

**Authors:** Kun Liang, Jinpeng Song, Rui Yuan, Zhizhou Ren

**Affiliations:** 1College of Artificial Intelligence, Nanjing Agricultural University, Nanjing 210031, China; 2College of Engineering, Nanjing Agricultural University, Nanjing 210031, China

**Keywords:** spectroscopic techniques, fusarium head blight, fingerprint region, data fusion

## Abstract

In this study, a method of mid-level data fusion with the fingerprint region was proposed, which was combined with the characteristic wavelengths that contain fingerprint information in NIR and FT-MIR spectra to detect the DON level in FHB wheat during wheat processing. NIR and FT-MIR raw spectroscopy data on normal wheat and FHB wheat were obtained in the experiment. MSC was used for pretreatment, and characteristic wavelengths were extracted by CARS, MGS and XLW. The variables that can effectively reflect fingerprint information were retained to build the mid-level data fusion matrix. LS-SVM and PLS-DA were applied to investigate the performance of the single spectroscopic model, mid-level data fusion model and mid-level data fusion with fingerprint information model, respectively. The experimental results show that mid-level data fusion with a fingerprint information strategy based on fused NIR and FT-MIR spectra represents an effective method for the classification of DON levels in FHB wheat samples.

## 1. Introduction

Wheat is one of the earliest cultivated crops in the world, with a large yield and high nutritional value. Approximately 40% of the global population depends on wheat as a dietary staple. Thus, wheat yield and security are crucial for food security. However, with global warming, Fusarium ear blight (scab), one of the three most notable wheat diseases, is seriously affecting wheat yield and quality. Wheat scab is considered a complex of cereal fungal diseases mostly caused by three types of fungi: *Fusarium culmorum*, *Fusarium graminearum* and *Fusarium avenaceum* [1]. The infection of wheat ears was shown to mainly occur in the flowering period, which can shrink the wheat grains, decrease the weight of single wheat grains and lead to a pale or pink grain color at maturity [2], resulting in a decrease in the wheat yield of approximately 10%. What is more important than the decreased yield is that the infected grains contain accumulated mycotoxins, such as deoxynivalenol (DON), which poses a serious threat to the health of people and livestock.

DON is highly toxic to animals and plants. The fungal toxin DON produced by the Fusarium pathogens of wheat scab can cause poisoning and lung and reproductive system infections, resulting in infertility, miscarriage and other conditions [3]. Developed countries worldwide have set very strict standards for the quarantine of Fusarium scab. If the toxin DON is detected in food, it cannot be consumed by humans. If the DON content exceeds 2 mg/kg, wheat grains cannot be purchased as feed. To protect humans from the risks related to these mycotoxins, it is urgent to control and detect wheat scab accurately and efficiently. Conventionally, several methods have been developed to measure and estimate the levels of DON and its derivatives, and the most common methods are chromatography and immunochemical techniques, such as high-performance liquid chromatography (HPLC), thin-layer chromatography (TLC), enzyme-linked immunosorbent assay (ELISA) and gas chromatography (GC) [4]. However, some of the traditional detection methods normally not only require a precise experimental environment but also consume too much time and can be affected by human error and are therefore not suitable for large-scale wheat scab detection [5]. For instance, TLC is cumbersome, with poor accuracy and specificity, and ELISA requires several reagents, produces a relatively large amount of waste and is known to yield biased detection results. 

To predict the disease level of wheat scab more accurately, infrared (IR) spectroscopy has been applied. In recent years, IR spectroscopy has become one of the most promising analytical tools available to researchers. This technique is associated with chemometrics and has been widely used in material identification and quantitative analysis because of its ability to achieve rapid and nondestructive detection with low energy consumption, simplicity and low costs [6].

Near-infrared (NIR) spectra contain information about overtone and combination bands of fundamental vibrations, and they are sensitive to many chemical groups and various molecular interactions [7,8]. NIR spectroscopy techniques have been widely applied in food composition determination and classification [9,10]. In recent decades, NIR spectroscopy has been widely used in the wheat industry for the determination of mycotoxins in wheat [11,12,13,14]. Using partial least-squares (PLS) regression and linear discriminant analysis (LDA), the authors showed that the overall classification and false complaint rates for the two models were 75–90% and 3–7%, respectively, and these studies proved that NIR techniques were suitable for detecting DON contamination in wheat grain [14]. In another article, modeling was performed using the soft independent modeling of class analogy (SIMCA) approach. The model was used to classify sound, scab-damaged and mold-damaged wheat kernels [15]. Levasseur-Garcia deduced that artificial neural networks achieved 98.8% successful identification of Fusarium by NIR spectroscopy [12]. The above studies all proved the feasibility of extrapolating wheat scab damage by NIR with high accuracy through different models.

Spectra in the mid-infrared MIR region (4000–400 ${cm}^{-1}$), obtained using Fourier-transform mid-infrared (FT-MIR) spectroscopy, provide information on the frequencies of fundamental molecular vibrations [16]. Molecules such as proteins and fatty acids can then be identified based on the FT-MIR spectral signals produced by these vibrations, which typically result in multiple and convoluted bands representing one molecule [17]. Currently, FT-MIR spectroscopy has been successfully applied to agricultural products [18,19,20]. Peiris et al. [21] used mid-infrared attenuated total reflection (Mid-IR-ATR) spectra (4000–380 ${cm}^{-1}$) to analyze Fusarium damage of wheat kernels caused by F. graminearum and DON contamination. They discovered a shift in the absorption peak and absorption of some bands that increased due to Fusarium damage of wheat kernels in the pericarp and germ. These phenomena indicate that MIR spectroscopy could detect the high concentration of DON that exists in Fusarium-damaged wheat kernels. Gregor Kos et al. [22] proposed a model to quickly detect corn infected with Fusarium fungi by attenuated total reflection Fourier-transform mid-infrared (FT-MIR-ATR) spectroscopy. When ergosterol and DON levels were more than 8.23 mg/kg and 0.13 mg/kg, respectively, the recognition accuracy exceeded 75%. This result proved the potential of the FT-MIR-ATR spectroscopic technique to identify corn that is infected with Fusarium fungi.

In addition, in the United States, Canada and some European countries, wheat quality requirements are relatively strict, and the wheat grain DON toxin content must be less than 0.5~2 ppm [23]. However, the concentration of DON in infected wheat is very low, even if it is substandard. Through experiments, it was found that it was difficult for IR spectroscopy to reflect and distinguish the change in the DON content in a specific band. However, with the aggravation of wheat scab, the luster and quality of wheat seeds change considerably. The contents of crude protein, cellulose and hemicellulose in wheat grains decrease, but the contents of reducing sugars and starch fat increase. There are corresponding bands in the IR spectrum related to the content information of these substances, so the degree of scab impact on wheat can be reflected by extracting and analyzing the characteristic bands of IR spectra.

The approach of combining spectroscopic data and data fusion plays a key role in geographical traceability, safety monitoring and quality control of agricultural products. This method could obtain more comprehensive information than spectral analysis alone and generally achieve better classification and prediction results because of the synergistic effect between different data [24]. Data fusion is divided into three levels: low-level, mid-level and high-level fusion [25]. Low-level data fusion is a simple method that rearranges data into a new matrix for combining inputs. While low-level data fusion offers simplicity, it can be time-consuming and computationally demanding due to the incorporation of varying proportions of predictive and irrelevant variables in the dataset [26]. Mid-level fusion effectively utilizes dimension-reduction techniques to eliminate unnecessary information, reduce noise, and extract relevant data, resulting in more efficient and accurate outcomes with reduced computational time [27]. High-level fusion involves combining the results obtained from models built using individual source data. It is generally more effective but requires large amounts of data and can be challenging to implement [28]. Stefano Schiavone et al. [29] used the low-level fusion method to distinguish pure and adulterated Grappa spirit, and the classification accuracy reached 100%. Sen Yao et al. [30] used the mid-level data fusion method of FT-IR and ultraviolet (UV) spectroscopies to establish the origin identification model of seven Boletus mushrooms. Yang Li et al. [31] combined spectral analysis with a high-level fusion strategy for the quantitative analysis of syrup added to honey products.

The aim of this study was to compare the selected NIR and FT-MIR spectral characteristic wavelengths with fingerprint information, and mid-level data fusion with the fingerprint information method was proposed to establish a classification model of Fusarium head blight (FHB) wheat. The effectiveness was proven by the comparison of model results between the mid-level data fusion with fingerprint information method, the single spectrum method and the mid-level data fusion method. The specific objectives were to (1) obtain raw data of different DON levels in FHB wheat by using NIR and the FT-MIR spectroscopic technique; (2) select the characteristic bands by using the CARS, MGS and XLW algorithms after the preprocessing of MSC and optimize the selection of characteristic bands combined with fingerprint information and (3) build classification models based on the PLS-DA and LS-SVM algorithms by mid-level data fusion with the fingerprint information method, the single spectrum method and the mid-level data fusion method and compare these model results to confirm the influence of the characteristic wavelength with fingerprint information fusion on the model accuracy. 

## 2. Materials and Methods

### 2.1. Sample Preparation

Wheat samples (Jimai 22) were collected from the Institute of Food Inspection of Jiangsu Academy of Agricultural Sciences (Nanjing, China). Dry pure potassium bromide was obtained from Nanjing Wanqing Chemical Glassware Instrument Co., Ltd. (Nanjing, China). Wheat samples were separated from all the stones, straw, grass seed, and soil clumps. The water content of wheat samples was measured directly through a non-destructive moisture meter (Model PM8188A, Taizhou Weikete Instrument Co., Ltd., Taizhou, China). The wheat samples with a high moisture content were ventilated and dried, while those with a low moisture content were exposed to moist air. Finally, the moisture content of wheat samples was controlled at 12–13% to prevent them from deteriorating or germinating before processing [32]. Moreover, a total of 120 wheat samples weighing 25 g each were numbered consecutively. Then, the samples were ground into a 20-mesh sieve by a solid sample mill (Changzhou Yuexin Instrument Manufacturing Co., Ltd., Changzhou, China). Milling allows for the reduction of microscopic air pockets, improves spectral quality and ensures the uniformity of wheat samples, thereby ensuring the stability and reliability of measurement results [33]. The wheat powder was mixed well and directly measured (NIR spectroscopy). Each wheat flour sample weighed 0.002 g and was used for tablet pressing and analysis (FT-MIR spectroscopy).

### 2.2. NIR and FT-MIR Spectra Acquisition

The NIR spectrometer (Infraxact, FOSS company, Hilleroed, Denmark) in the spectral region ranging from 570 to 1848 nm was used for collecting NIR spectra in reflectance mode. Spectra were acquired with a 2 nm spectral resolution. The instrument was recalibrated after warming up for 30 min to ensure the accuracy and reliability of the measurement data.

FT-MIR spectra were collected at room temperature with an FTIR spectrometer (Nicolet iS10, Thermo Fisher Company, Waltham, MA, USA). The spectrum of the pure potassium bromide tablet was used as background.

Preparation of wheat sample tablets: The potassium bromide used in this paper were stored in a light-protected glass container. Prior to use, potassium bromide was baked in an oven (LC-202, Shanghai Lichen Instrument Technology Co., Ltd., Shanghai, China) at 120 °C for 24 h before use to remove moisture. A total of 0.2 g of dried pure potassium bromide was ground in an agate mortar until the powder stuck to the mortar wall. Then, 0.002 g of wheat flour sample was added, mixed thoroughly with the potassium bromide while grinding and finally put into a tablet press machine (YP-2, Shanghai Shanyue Instrument Co., Ltd., Shanghai, China) and pressed into sheets. The wheat sample tablets were loaded into the sample pool and scanned. The range used for collecting FT-MIR spectra was from 4000 to 400 ${cm}^{-1}$. Spectra were acquired with a 4 ${cm}^{-1}$ spectral resolution. Each spectrum recorded by the instrument was an average of 16 scans.

### 2.3. HPLC Determination of DON Content Level

According to the SN/T3137-2012 standard (Shaanxi Entry-Exit Inspection and Quarantine Bureau and Shenzhen Entry-Exit Inspection and Quarantine Bureau, Shaanxi and Shenzhen, China), the DON content was determined quantitatively using liquid chromatography–mass spectrometry (LC–MS). The extraction solution comprised acetonitrile water (84:16), 3 mL of which was processed per sample at a centrifugal speed of 2500 rpm, and the extracted sample was then passed through a solid-phase extraction cartridge at a flow rate of 1 mL/min. The LC–MS system (3500 QTRAP, ABSCIEX, Redwood City, CA, USA) was used to determine toxins with the injection volume set to 5 mL, and the content limit for DON was set to 20 μg/kg. Mobile phase A consisted of 5 mmol/L ammonium acetate in water, and at 0, 3, 7, 13, 13.1 and 16 min, mobile phase A and mobile phase B were at ratios of 85:15, 30:70, 20:80, 10:90, 85:15 and 85:15, respectively, and the flow rate was set to 0.6 mL/min.

### 2.4. Data Analysis

#### 2.4.1. Spectral Pretreatment

IR spectra were subjected to data processing to remove spectral changes that may be unrelated to chemical composition. Spectral data preprocessing was accomplished using OMNIC 8.0 software (Thermo Nicolet, Minneapolis, MN, USA). FT-MIR spectral data were imported into OMNIC software and subtracted from the atmospheric background and baseline corrected, and the transmittance at each wavenumber of each sample was taken as the spectral data of the sample. Both the FT-MIR spectrum after processing and the raw NIR spectrum were analyzed by multiple scattering correction (MSC). The MSC algorithm is often used to eliminate the scattering effect caused by the nonuniformity of solid samples or the size of the particles, and it is also used in the NIR diffuse reflectance spectrum of solids and turbid liquids.

#### 2.4.2. Feature Extraction and Data Fusion

The use of the whole spectral region from vibrational spectroscopy does not always result in a good multivariate prediction model, as many of the wavenumbers do not contain the necessary information, which causes interference in the model [6]. The selection of features was helpful to reduce redundancy and collinearity and improve the performance of the correction model.

The features were selected by three feature extraction algorithms, including competitive adaptive reweighted sampling (CARS), modified Gram–Schmidt (MGS), and X-loading weights (XLW). CARS is a characteristic wavelength technique based on the principle of simulating Darwinian evolution (the survival of the fittest) and was proposed in 2009 [34]. In the process of performing the CARS calculation, a partial least-squares regression model was established by random sampling (Monte Carlo sampling, MCS) and then filtering wavelength variables based on adaptive reweighted sampling (ARS) and exponentially decreasing function (EDP). Finally, the combination of optional wavelength variables with the smallest root mean square error of cross-validation (RMSECV) was obtained from all sampled variable sets through cross-validation. MGS is helpful in selecting the key wavelengths among the infrared spectrum after pretreatment. The basic idea of MGS is based on the projection principle, a new orthogonal basis constructed based on the original. XLW can be used as an algorithm to extract characteristic wavelengths. XLW is based on the modeling result of partial least-squares (PLS) analysis [35]. Each latent variable (LV) was generated by the PLS modeling results, and the regression coefficients corresponding to each wavelength point were obtained. The absolute regression coefficients were used to check the effects of different wavelengths on the prediction accuracy of the model built. Therefore, the characteristic wavelength can be extracted according to the absolute value of the load coefficient corresponding to each wavelength under an LV [36]. Large regression coefficient values indicated the importance and significance of the characteristic wavelength corresponding to the LV.

To obtain a more accurate model with less redundant and more effective information, a mid-level data fusion strategy with fingerprint information is proposed in this paper. The critical process of this strategy is the method of constructing the dataset. There are two types of variables that are selected into the mid-level data fusion matrix: one is the characteristic wavelengths and wavenumbers in the regions with obvious absorbance differences of different DON levels in FHB wheat, and the other is the wavenumbers and wavelengths in accordance with the fingerprint information.

### 2.5. DON Levels Classification

The partial least-squares discriminant analysis (PLS-DA) and least-squares support vector machine (LS-SVM) classification models of DON levels in FHB wheat were established based on near-infrared and Fourier-transform mid-infrared spectroscopy, and characteristic wavelengths selected by the previous algorithms were taken as model input. The most effective classification model and technique of wheat infective scab grade was summarized by analyzing the recognition accuracy of different models. In this study, the classification vector was set as [1, 2], where 1 means that the DON content in the sample was lower than 1000 ppb, representing normal wheat, and 2 means that the DON content in the wheat sample was higher than 1000 ppb, representing fusarium head blight wheat. PLS-DA discrimination is a qualitative analysis method based on PLS regression. The principal component number is one of the main factors affecting the performance of the PLS-DA model. The optimal principal component number of this model was determined by applying the leave-one-out (LOO) cross-validation (CV) technique in this experiment. The LS-SVM method maps the input data from the conventional space into a higher dimensional space of function fitting. It considers equality-type constraints instead of inequality-type constraints, and the minimization loss function is solved in this higher dimensional space; namely, the quadratic programming is transformed into the linear programming problem, then the computational complexity is decreased, and the computing speed is improved greatly. The RBF kernel function was chosen as the kernel function of the LS-SVM model in the establishment of DON levels in the FHB wheat classification model based on LS-SVM, in which parameters γ and δ2 were automatically adjusted by 10-fold cross-validation.

### 2.6. Software

All the data processing and model algorithms were realized in MATLAB (version R2020ba, MathWorks, Natick, MA, USA). Functions in MATLAB were used to calculate the preprocessing of NIR and FT-MIR data as well as extract latent variables and establish PLS-DA and LS-SVM models.

## 3. Results and Discussion

### 3.1. Spectral Analysis and Wavelength Selection

#### 3.1.1. Raw Spectral Analysis and Spectral Pretreatment

Figure 1 shows the raw spectra and spectra that were pretreated using MSC. The absorbance of the two sample types of DON levels in FHB wheat was different in different bands, but the overall trend was consistent. In the NIR spectrum, the wavelength region between 570 nm and 780 nm is related to the epidermal gloss of wheat samples. The absorption peak at approximately 1000 nm is mainly caused by the second overtones of O-H and N-H in the protein [37]. The band related to water moisture is located at 1125–1152 nm, which is caused by the combination of the first overtone of the O-H stretching band and the O-H bending band [38]. The peak at approximately 1210 nm is also related to moisture information [39]. The absorption at 1360 nm is produced by the combined frequency band of aliphatic hydrocarbon C-H and is related to lipid information. The spectral region near 1730 nm linked to the vibration of amylose reflects starch information [40]. In the FT-MIR spectra, the C-O stretching vibration absorption peak of polysaccharides in carbohydrates is approximately 1000 ${cm}^{-1}$. The absorption peak at 1550–1570 ${cm}^{-1}$ is mainly caused by the N-H bending vibration and C-N stretching vibration in the protein and the antisymmetric stretching vibration of the aliphatic nitro compound NO_2_. The intensity of the N-H bending vibration peak reflects the protein content in wheat. The absorption peak at 2800–3000 ${cm}^{-1}$ is due to C-H stretching vibrations in starch [41]. The absorption peak at 1600–1700 ${cm}^{-1}$ belongs to the amide I region, which is related to protein [42]. After applying the MSC algorithm for preprocessing, the noise in the raw reflectance spectra data of NIR and FT-MIR is reduced, which is caused by light scattering and baseline drift.

#### 3.1.2. Characteristic Wavelength Selection

The results for the three feature extraction algorithms of MGS, CARS, and XLW are shown in Figure 2. In the NIR spectrum, all three algorithms selected characteristic wavelengths of approximately 570–780 nm, which reflect the epidermal gloss of wheat samples. From Figure 2a, thirteen characteristic wavelengths (584 nm, 592 nm, 606 nm, 610 nm, 624 nm, 670 nm, 680 nm, 698 nm, 700 nm, 970 nm, 1100 nm, 1102 nm and 1118 nm) were selected by MGS and concentrated at 570–780 nm, reflecting the epidermal gloss of wheat samples. In addition, the characteristic wavelength near 1000 nm is associated with protein information, and the three characteristic wavelengths of approximately 1125–1152 nm correspond to moisture information. Nine characteristic wavelengths (676 nm, 678 nm, 752 nm, 754 nm, 1018 nm, 1192 nm, 1306 nm, 1620 nm, 1716 nm) were selected by CARS. Among them, the characteristic wavelength at approximately 1000 nm is relevant to protein information, the characteristic wavelength at approximately 1210 nm is correlated with moisture information, the characteristic wavelength at approximately 1306 nm reflects fat content information and the two characteristic wavelengths at approximately 1730 nm correspond to starch information. After applying XLW, five variables (576 nm, 570 nm, 672 nm, 674 nm and 1326 nm) were determined as the characteristic wavelength subset, and the characteristic wavelength of approximately 1306 nm is related to fat content information. In the FT-MIR spectrum, it is observed from Figure 2b that thirteen characteristic wavenumbers (1179.307 ${cm}^{-1}$, 1196.664 ${cm}^{-1}$, 1545.732 ${cm}^{-1}$, 1569.839 ${cm}^{-1}$, 2391.4 ${cm}^{-1}$, 2396.221 ${cm}^{-1}$, 2401.043 ${cm}^{-1}$, 2405.864 ${cm}^{-1}$, 2410.686 ${cm}^{-1}$, 2432.864 ${cm}^{-1}$, 2467.578 ${cm}^{-1}$, 2481.078 ${cm}^{-1}$, 2519.649 ${cm}^{-1}$) were selected by MGS. The two characteristic wavenumbers of absorption peaks at approximately 1550–1570 ${cm}^{-1}$ are mainly associated with the information of protein and fat. Thirteen characteristic wavenumbers (1174.486 ${cm}^{-1}$, 1175.45 ${cm}^{-1}$, 1176.415 ${cm}^{-1}$, 1525.482 ${cm}^{-1}$, 1526.446 ${cm}^{-1}$, 1527.41 ${cm}^{-1}$, 1528.375 ${cm}^{-1}$, 1530.303 ${cm}^{-1}$, 1531.267 ${cm}^{-1}$, 1532.232 ${cm}^{-1}$, 1645.052 ${cm}^{-1}$, 2859.073 ${cm}^{-1}$, 3671.957 ${cm}^{-1}$) were selected by CARS. Its characteristic wavenumbers located at the absorption peak at approximately 1550–1570 ${cm}^{-1}$ and 1600–1700 ${cm}^{-1}$ mainly reflect protein and fat information. Based on the XLW method, nine characteristic wavenumbers (400.1738 ${cm}^{-1}$, 413.6736 ${cm}^{-1}$, 817.7045 ${cm}^{-1}$, 1025.024 ${cm}^{-1}$, 1026.952 ${cm}^{-1}$, 1839.835 ${cm}^{-1}$, 2359.579 ${cm}^{-1}$, 2891.859 ${cm}^{-1}$ and 2868.716 ${cm}^{-1}$) are selected, with absorption peaks at approximately 1000 ${cm}^{-1}$ and 2800–3000 ${cm}^{-1}$ mainly reflecting starch information.

#### 3.1.3. Fingerprint Information of Characteristic Wavelength Selected

From Figure 1a and Figure 2a, all the wavelengths selected in the NIR spectra by the three algorithms are located in the absorption peak regions, which are correlated with fingerprint information, and obvious differences in the absorbance of different DON levels in FHB wheat are reflected at these wavelengths. Therefore, all the characteristic wavelengths of the NIR spectra were selected for mid-level data fusion. However, in the FT-MIR spectra, the characteristic wavenumbers were selected in the regions with obvious absorbance differences of different DON levels in FHB wheat or in accordance with effectively the fingerprint information. Therefore, for the mid-level data fusion, four characteristic wavenumbers related to fingerprint information (1025.024 ${cm}^{-1}$, 1026.952 ${cm}^{-1}$, 2891.859 ${cm}^{-1}$, 2868.716 ${cm}^{-1}$) and three characteristic wavenumbers (400.1738 ${cm}^{-1}$, 413.6736 ${cm}^{-1}$, 817.7045 ${cm}^{-1}$) in the significantly different absorbance were selected in the XLW method in the FT-MIR spectra. Regarding the characteristic wavenumbers selected by the MGS method in the FT-MIR spectra, two wavenumbers (1545.732 ${cm}^{-1}$, 1569.839 ${cm}^{-1}$) associated with protein and fat information and the other two wavenumbers (1179.307 ${cm}^{-1}$, 1196.664 ${cm}^{-1}$) reflecting the obvious absorbance difference were used for the mid-level data fusion matrix. Five wavenumbers (1526.446 ${cm}^{-1}$, 1528.375 ${cm}^{-1}$, 1530.303 ${cm}^{-1},$ 1532.232 ${cm}^{-1}$, 1645.052 ${cm}^{-1}$) corresponding to the information of protein and fat and the other two wavenumbers (1174.486 ${cm}^{-1}$, 1176.415 ${cm}^{-1}$) located in the region with considerable absorbance difference were selected in the CARS method in the FT-MIR spectra. Then, the mid-level data fusion matrix was constructed by using the characteristic wavelengths in NIR spectra and the selected characteristic wavenumbers in FT-MIR spectra.

### 3.2. Model for NIR Data

Classification models using LS-SVM and PLS-DA were developed for DON levels in FHB wheat based on the selected NIR spectral wavelengths after different feature extraction algorithms. The accuracy rates of the training set and the test set are shown in Table 1. The accuracy of the LS-SVM and PLS-DA models is above 89%. The classification accuracy of the PLS-DA models based on the CARS and MGS methods outperformed that based on the XLW method, and the LS-SVM models exhibited similar performance. The reason may be that the wavelengths selected by the XLW method reflect limited information, while the models with the CARS and MGS methods contain more comprehensive information. The epidermal gloss information (570–780 nm) was selected by all three methods. For the XLW method, there is only one sensitive wavelength at 1326 nm to reflect fat content information except for the epidermal gloss information. However, the characteristic wavelengths selected by CARS correspond to the fat content information (1306 nm), moisture content (1018 nm and 1192 nm), and starch information (1716 nm). Regarding the MGS method, the protein information is reflected by the characteristic wavelength at 970 nm, and the moisture information is reflected by characteristic wavelengths at 1100 nm, 1102 nm and 1118 nm.

### 3.3. Model for FT-MIR Data

LS-SVM and PLS-DA were implemented to establish classification models of DON levels in FHB wheat based on the selected characteristic wavenumbers in FT-MIR spectra. The accuracy of the training set and test set are recorded in Table 1. Except for the XLW-PLSDA and XLW-LSSVM models, the accuracy of all the FT-MIR models was more than 80% but lower than the accuracy of the NIR models. The FT-MIR models showed lower prediction accuracy and abilities than the NIR models because the FT-MIR region was shown to have lower energy than the NIR region and is more easily affected by multiple interferences [42,43,44]. Similar to the models in the NIR region, the accuracy of the PLS-DA model established by three different feature extraction algorithms using FT-MIR spectroscopy was higher than that of the LS-SVM model. Additionally, similar to NIR models, the accuracy of the PLS-DA and LS-SVM models established by MGS was higher than that of the models established by CARS and XLW.

### 3.4. Mid-Level Data Fusion Model

The characteristic wavenumbers or wavelengths from FT-MIR and NIR spectra were applied for mid-level data fusion, and then the fusion data were used to establish the LS-SVM and PLS-DA classification models. The accuracy of the training set and the test set are shown in Table 1. The performance of the model established by the mid-level data fusion was improved to varying degrees compared with the model established by FT-MIR spectroscopy, with the accuracy of the training set and the test set increasing 6.25–14.05% and 3.45–17.25%, respectively. However, the classification accuracy of the model established by the mid-level data fusion data matrix was slightly lower than that of the model established by NIR spectroscopy. The accuracy of the training set and the test set of the mid-level fusion model decreased by 0.32–3.48% and 0–6.89%, respectively, compared with that of the model based on NIR spectroscopy. The exception was the accuracy of the training set of the XLW-LSSVM model, which increased by 0.88%. The reason was that mid-level data fusion added the characteristic wavelengths of NIR spectroscopy, thus supplementing the information and improving the accuracy of the model based on the mid-level data fusion matrix compared with that of the FT-MIR models. The lower accuracy of the mid-level data fusion model compared to the NIR model can be attributed to the presence of a larger number of irrelevant variables, which negatively affected the model performance. These irrelevant variables originated from the selected characteristics of the FT-MIR spectrum, leading to a decrease in classification accuracy for detecting DON levels defect of Fusarium head blight in wheat.

### 3.5. Mid-Level Data Fusion Model with Fingerprint Information

The characteristic wavelengths and wavenumbers that can effectively reflect fingerprint information are reselected from the variables selected by the three feature extraction algorithms. Subsequently, the mid-level data fusion matrix of the reselected wavelengths and wavenumbers was built, and then the matrix was taken as model input to establish the PLS-DA and LS-SVM models. The model’s performance is recorded in Table 1. All mid-level data fusion with fingerprint information models used fewer wavenumbers than the mid-level data fusion model. In this case, the number of the reselected wavelengths based on CARS decreased from 22 to 16 and that based on MGS and XLW decreased from 26 to 20 and from 14 to 12, respectively. Moreover, the accuracy of almost every model was improved, with the accuracy of the training set and the test set increasing 0.43–3.49% and 0–10.34%, respectively. This result suggests that mid-level data fusion with fingerprint information can reduce complexity, effectively reflect fingerprint information and improve model accuracy. The accuracy of the training set and the test set of some models based on mid-level data fusion with fingerprint information increased 0.01–2.04% and 0–3.45%, respectively, compared with the NIR-based model. The accuracy of the training set and the test set of the mid-level data fusion with fingerprint information model increased 7.37–15.21% and 13.79–20.69%, respectively, compared with the FT-MIR-based model. The reason for the improved classification performance may be that after selection, the characteristic wavenumbers that can effectively reflect fingerprint information were retained, reducing the impact of irrelevant information. Benefiting from data fusion, complementary NIR and FT-MIR information was obtained, and a more accurate model with less redundant information was established. Based on the mid-level data fusion with fingerprint information strategy, the best classification performance was obtained by the MGS-PLS-DA model, with the accuracy of the training set and test set reaching 99% and 96.55%, respectively. This model was established by combining the characteristic wavelengths that contain the fingerprint information of epidermal gloss, protein content, and moisture content in NIR spectra and wavenumbers that contain the protein and fat information in FT-MIR spectra.

## 4. Conclusions

In this study, a method of mid-level data fusion with fingerprint information in NIR and FT-MIR spectroscopy was proposed to classify two FHB wheat samples with two DON levels. The raw spectral data of different DON levels in FHB wheat were obtained based on NIR and FT-MIR spectroscopy. Then, the obtained spectral data were analyzed by MSC for pretreatment, and features were extracted by CARS, MGS and XLW. To establish more concise and efficient models, the characteristic wavenumbers and wavelengths that can effectively reflect fingerprint information were retained to reduce the impact of irrelevant information after feature selection. Subsequently, the mid-level data fusion matrix of the reselected wavelengths and wavenumbers was constructed. The PLS-DA and LS-SVM algorithms were used to establish mid-level data fusion with a fingerprint information model, mid-level data fusion model and single spectrum models. It was noted that compared with the other three types of models, the accuracy of the mid-level data fusion with fingerprint information model increased to varying degrees (the accuracy of the training set and the test set increased by 0.01–14.62% and 0–20.69%, respectively). Compared with the mid-level data fusion strategy, the strategy of mid-level data fusion with fingerprint information selected fewer characteristic wavelengths and wavenumbers. The number of input variables from the model based on CARS decreased from 22 to 16, and that based on MGS and XLW decreased from 26 to 20 and from 14 to 12, respectively. Hence, more precise models using fewer characteristic wavelengths and wavenumbers were obtained through mid-level data fusion with a fingerprint information strategy. Finally, the optimal model was determined to be the MGS-PLSDA model based on mid-level data fusion with a fingerprint information strategy. The optimal rate of the training set was 99%, and that of the test set was 96.55%. This paper introduces the novel strategy of mid-level data fusion with fingerprint information, which selects fewer characteristic wavelengths and wavenumbers compared to traditional mid-level data fusion. The model built using this strategy exhibits improved classification performance while being more concise and efficient, reducing computational costs. This highlights the importance of incorporating fingerprint information in data fusion models and provides valuable insights for researchers and practitioners in various fields. Additionally, the model enables accurate classification of DON defects in Fusarium head blight wheat, offering a new method for agricultural testing and contributing to crop quality and yield improvement.

## Figures and Tables

**Figure 1 sensors-23-06600-f001:**
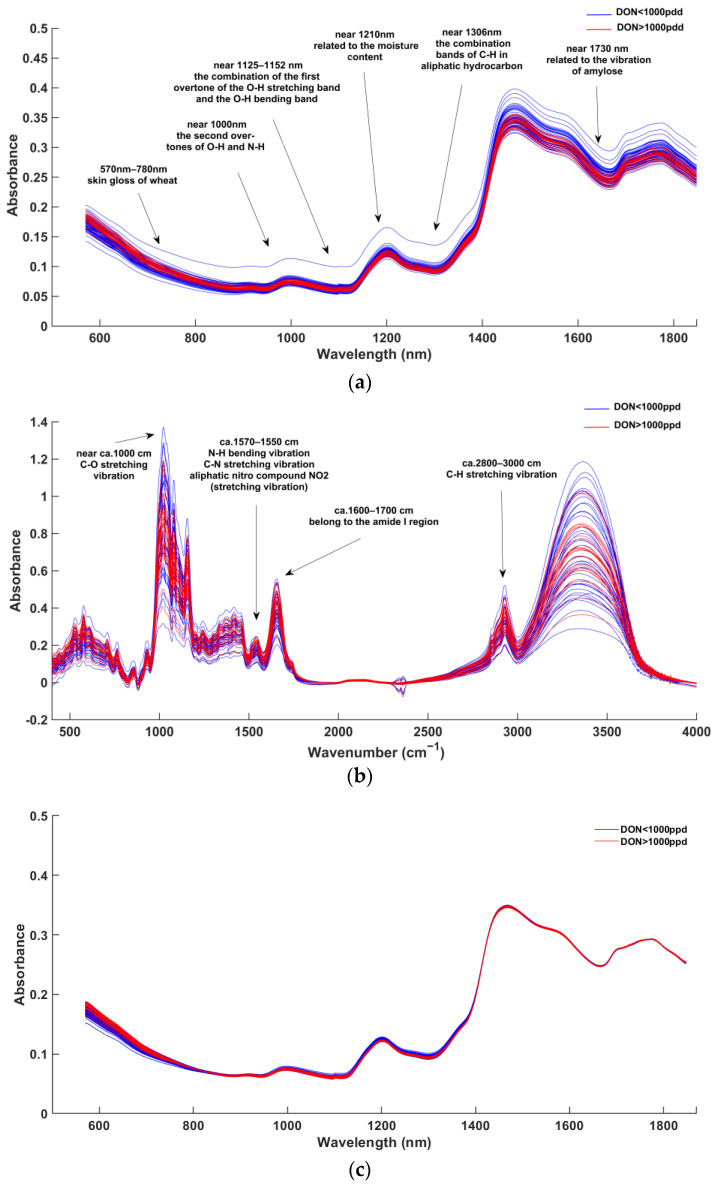
Shows the raw spectra of wheat samples of NIR (**a**) and FT-MIR (**b**); the spectra were pretreated using MSC of wheat samples of NIR (**c**) and FT-MIR (**d**).

**Figure 2 sensors-23-06600-f002:**
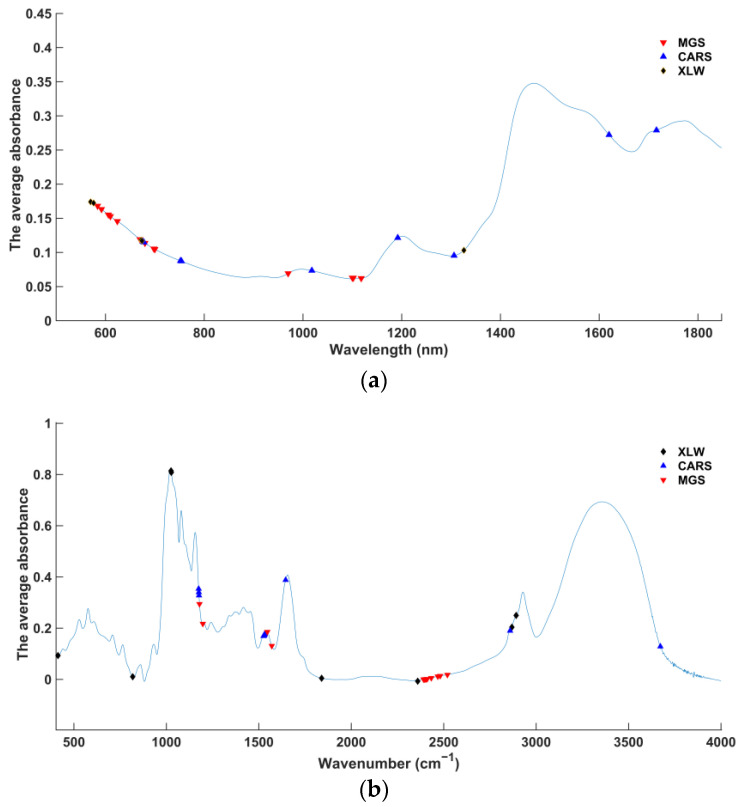
Selected variables (**a**) Average NIR spectra. (**b**) Average FT-MIR spectra.

**Table 1 sensors-23-06600-t001:** This is a table. Results of classification on accuracy of PLS-DA and LS-SVM modeling with the different data fusion strategies.

Model	NIR			FTIR			Mid-Level Data Fusion			Fingerprint Information for Mid-Level Data Fusion		
	Number of Input Variable	Training Set (%)	Test Set (%)	Number of Input Variable	Training Set (%)	Test Set (%)	Number of Input Variable	Training Set (%)	Test Set (%)	Number of Input Variable	Training Set (%)	Test Set (%)
CARS-PLSDA	9	96.25	93.10	13	87.94	82.76	22	94.19	86.21	16	97.03	96.55
MGS-PLSDA	13	98.89	96.55	13	91.63	86.26	26	98.57	93.10	20	99.00	96.55
XLW-PLSDA	5	93.49	93.10	9	79.21	72.41	14	90.56	89.66	12	93.83	93.10
CARS-LSSVM	9	97.32	96.55	13	86.00	82.76	22	93.84	93.10	16	97.33	96.55
MGS-LSSVM	13	97.09	96.55	13	88.50	82.76	26	95.84	93.10	20	98.75	96.55
XLW-LSSVM	5	90.10	89.66	9	76.93	75.86	14	90.98	89.66	12	92.14	89.66

## Data Availability

Not applicable.

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
