# Peer review of "Mid-Level Data Fusion Combined with the Fingerprint Region for Classification DON Levels Defect of Fusarium Head Blight Wheat"

_sensors, 2023, doi:10.3390/s23146600_

Round 1
Reviewer 1 Report
The study is important for the development of food safety diagnostic systems. The topic is important and relevant.
- in the research methods section, I would like to see clearly how the potassium bromide used to make the tablets was kept constantly dry, and also how the tablets were stored, since potassium bromide is very hydrophilic.
- when discussing the results, I would like to see how the data obtained by the authors and the proposed models correlate with the data and models obtained by other researchers (if any). Based on the data obtained, it is necessary to show the novelty of the proposed approach more clearly.
Author Response
Dear Editor and reviewers,
Thank you for your letter dated Jun 13, 2023. We are grateful for the suggestions, which have enabled us to improve our work. We have uploaded the file of the revised manuscript, and major revisions are marked in red. We appreciate the detailed and useful comments and suggestions.
Appended to this letter are our point-by-point responses to the comments. The comments have been reproduced, and our responses are given directly afterward in red.
Thank you again for your time and your advice, and we hope to learn more from you.
Sincerely,
Kun Liang
Reviewer 1:
The study is important for the development of food safety diagnostic systems. The topic is important and relevant.
- in the research methods section, I would like to see clearly how the potassium bromide used to make the tablets was kept constantly dry, and also how the tablets were stored, since potassium bromide is very hydrophilic.
when discussing the results, I would like to see how the data obtained by the authors and the proposed models correlate with the data and models obtained by other researchers (if any). Based on the data obtained, it is necessary to show the novelty of the proposed approach more clearly.
Q1. in the research methods section, I would like to see clearly how the potassium bromide used to make the tablets was kept constantly dry, and also how the tablets were stored, since potassium bromide is very hydrophilic.
Thank you for the suggestion. We have added the relevant explanation in the revised manuscript (Lines 161-164) as follows:“The potassium bromide used in this paper should be stored in a light-protected glass container. Prior to use, potassium bromide needs to be baked in an oven (LC-202, Shanghai Lichen Instrument Technology Co., LTD, Shanghai, China) at 120°C for 24 hours before use to remove moisture.”
Q2. when discussing the results, I would like to see how the data obtained by the authors and the proposed models correlate with the data and models obtained by other researchers (if any). Based on the data obtained, it is necessary to show the novelty of the proposed approach more clearly.
Answer: Thank you for the suggestion. We have identified similar works by other researchers and compared them with our proposed approach based on the data obtained. The results of the comparison are as follows: The linear discriminant analysis model proposed by the He et al. [1] incorporates both spectral and texture features, achieving accuracies of 95.06% for the training set and 91.36% for the test set. In this study, the optimal model, determined to be the MGS-PLSDA model based on mid-level data fusion with a fingerprint information strategy, outperformed the previous model with a training set accuracy of 99% and a test set accuracy of 96.55%. This indicates that the optimal model achieved superior classification performance, further highlighting the effectiveness of the proposed mid-level data fusion approach with fingerprint information.”
The above content will not be included in the manuscript.
[1] He, X., Zhao, T., Shen, F., Liu, Q., Fang, Y., & Hu, Q. (2021). Online detection of naturally DON contaminated wheat grains from China using Vis-NIR spectroscopy and computer vision. Biosystems Engineering.
Reviewer 2 Report
This study provides a method of mid-level data fusion with the fingerprint region. The structure of the proposed paper is clear which is easy to read and understand, but there are still some points which should be improved.
1 What is the advantage in view to other data fusion including low-level fusion and high-level fusion?
2 The effect of mid-level data fusion (line 375,compared with the model based on NIR, the accuracy of the proposed model in this study is reduced).
3 The study did not provide a comparison of the effect before and after spectral MSC preprocessing algorithm.
4 Please complete the sample preparation – images of wheat samples.
5 What is the effect of fingerprint information in classifying two FHB wheat samples?
6 Did you test different algorithms other than the PLS- DA and LS-SVM algorithms?
7 What is new? What is the scientific impact of this work? What’s about the advantage for agriculture? This should be pointed out!
Please check English since there are some typos and grammatical problems, I will suggest the authors to make all corrections.
Author Response
Dear Editor and reviewers,
Thank you for your letter dated Jun 13, 2023. We are grateful for the suggestions, which have enabled us to improve our work. We have uploaded the file of the revised manuscript, and major revisions are marked in red. We appreciate the detailed and useful comments and suggestions.
Appended to this letter are our point-by-point responses to the comments. The comments have been reproduced, and our responses are given directly afterward in red.
Thank you again for your time and your advice, and we hope to learn more from you.
Sincerely,
Kun Liang
Reviewer 2
This study provides a method of mid-level data fusion with the fingerprint region. The structure of the proposed paper is clear which is easy to read and understand, but there are still some points which should be improved.
Q1. What is the advantage in view to other data fusion including low-level fusion and high-level fusion?
Answer: Thank you for the suggestion. We have added the relevant explanation in the revised manuscript (Lines 106-115) as follows:
“Low-level data fusion is a simple method that rearranges data into a new matrix for combining inputs. While low-level data fusion offers simplicity, it can be time-consuming and computationally demanding due to the incorporation of varying proportions of predictive and irrelevant variables in the dataset [26]. Mid-level fusion effectively utilizes dimension reduction techniques to eliminate unnecessary information, reduce noise, and extract relevant data, resulting in more efficient and accurate outcomes with reduced computational time. [27]. High-level fusion involves combining the results obtained from models built using individual source data. It is generally more effective, but requires large amounts of data and can be challenging to implement. [28].”
[26] Casian. T, Nagy. B, Kovács. B, Galata. D, Hirsch. E, Farkas. A., Molecules., 27:15 (2022). https://doi.org/10.3390/molecules27154846
[27] Hayes. E, Greene. D, O'Donnell. C, O'Shea. N, Fenelon. MA., Frontiers in Nutrition. 9:1074688 (2022). https://doi.org/10.3389/fnut.2022.1074688.
[28] Chen. J, Ma. J, Han. X, Zhou. Y, Xie. B, Huang. F, Li. L, Li. Y., Journal of Biophotonics. 16, (3), (2023). https://doi.org/10.1002/jbio.202200251
Q2. The effect of mid-level data fusion (line 375,compared with the model based on NIR, the accuracy of the proposed model in this study is reduced).
Answer: Thank you for the suggestion. Due to the lower energy of the FT-MIR spectrum, it is susceptible to various factors and may contain interfering information. We have included the relevant explanation in revised the manuscript (Lines 402-407) as follows:“The lower accuracy of the mid-level data fusion model compared to the NIR model can be attributed to the presence of a larger number of irrelevant variables, which negatively affected the model performance. These irrelevant variables originated from the selected characteristics of the FT-MIR spectrum, leading to a decrease in classification accuracy for detecting DON levels defect of Fusarium head blight in wheat.”
Q3. The study did not provide a comparison of the effect before and after spectral MSC preprocessing algorithm.
Answer: Thank you for the suggestion. We have added Figure 1(c) illustrating the NIR spectral curve after preprocessing with the MSC algorithm, as well as Figure 1(d) displaying the FT-MIR spectral curve after preprocessing with the MSC algorithm. In the revised manuscript (Lines 280-282), detailed explanations have been provided for these images as follows:“After applying the MSC algorithm for preprocessing, the noise in the raw reflectance spectra data of NIR and FT-MIR is reduced, which is caused by light scattering and baseline drift.”
Please see Figure 1. in the revised manuscript
Figure 1. Shows the raw spectra of wheat samples of NIR (a) and FT-MIR (b), the spectrum were pretreated using MSC of wheat samples of NIR (c) and FT-MIR (d).
Q4. Please complete the sample preparation – images of wheat samples.
Answer: Thank you for the suggestion. The samples are the wheat samples after milling, and with the normal milling samples no difference, so there were no pictures in the paper.
Q5. What is the effect of fingerprint information in classifying two FHB wheat samples?
Answer: Thank you for the suggestion. Fingerprint information can reflect the differences in protein, fat, moisture, and skin color between two types of FHB wheat samples, thus enabling effective classification of the two types of FHB wheat samples.
The following explanation was originally provided in lines 411-414 of the original manuscript. It is now located in lines 433-436 of the revised manuscript, as follows: “This model was established by combining the characteristic wavelengths that contain the fingerprint information of epidermal gloss, protein content, and moisture content in NIR spectra and wavenumbers that contain the protein and fat information in FT-MIR spectra.”
Q6. Did you test different algorithms other than the PLS- DA and LS-SVM algorithms?
Answer: Thank you for the suggestion. This paper did not attempt other algorithms but selected the classical linear model PLS-DA and the classical non-linear model SVM for modeling. The purpose was to demonstrate the effectiveness of method of mid-level data fusion with fingerprint information.
Q7. What is new? What is the scientific impact of this work? What’s about the advantage for agriculture? This should be pointed out!
Answer: Thank you for the suggestion. We have added the relevant explanation in the revised manuscript (Lines 473-481) as follows:
“This paper introduces the novel strategy of mid-level data fusion with fingerprint information, which selects fewer characteristic wavelengths and wavenumbers compared to traditional mid-level data fusion. The model built using this strategy exhibits improved classification performance while being more concise and efficient, reducing computational costs. This highlights the importance of incorporating fingerprint information in data fusion models and provides valuable insights for researchers and practitioners in various fields. Additionally, the model enables accurate classification of DON defects in Fusarium head blight wheat, offering a new method for agricultural testing and contributing to crop quality and yield improvement.”

Round 2
Reviewer 2 Report
The authors have revised the manuscript according to the comments. Lot of work has been done. The revised manuscript is better and clearly written. However, I think it would be helpful to provide further details about the milling process and how it may have affected the samples.
I would suggest minor revising.
The authors have revised the manuscript according to the comments. Lot of work has been done. The revised manuscript is better and clearly written. However, I think it would be helpful to provide further details about the milling process and how it may have affected the samples.
I would suggest minor revising.
Author Response
Dear Editor and reviewers,
Thank you for your letter dated Jun 29, 2023. We are grateful for the suggestions, which have enabled us to improve our work. We have uploaded the file of the revised manuscript, and major revisions are marked in red. We appreciate the detailed and useful comments and suggestions.
Appended to this letter are our point-by-point responses to the comments. The comments have been reproduced, and our responses are given directly afterward in red.
Thank you again for your time and your advice, and we hope to learn more from you.
Sincerely,
Kun Liang
Reviewer 2:
The authors have revised the manuscript according to the comments. Lot of work has been done. The revised manuscript is better and clearly written. However, I think it would be helpful to provide further details about the milling process and how it may have affected the samples.
I would suggest minor revising.
Thank you for the suggestion. We have added the relevant explanation in the revised manuscript (Lines 149-151) as follows:
Milling allows for the reduction of microscopic air pockets, improves spectral quality, and ensures the uniformity of wheat samples, thereby ensuring the stability and reliability of measurement results [33].
[33] C.R. Whatley, N.K. Wijewardane, R. Bheemanahalli, K.R. Reddy, Y. Lu., Effects of fine grinding on mid-infrared spectroscopic analysis of plant leaf nutrient content, Scientific Reports, 18;13(1):6314 (2023). https://doi.org/10.1038/s41598-023-33558-5.
Furthermore, we will present relevant pictures of the sample preparation process, but these images will not be included in the revised manuscript.
